# Disruption of the *Pseudomonas aeruginosa* Tat system perturbs PQS-dependent quorum sensing and biofilm maturation through lack of the Rieske cytochrome $bc_1$ sub-unit

Eliza Ye-Chen Soh[1‡], Frances Smith[1‡], Maxime Rémi Gimenez[2], Liang Yang[3¤], Rebecca Munk Vejborg[4], Matthew Fletcher[1], Nigel Halliday[1], Sophie Bleves[2], Stephan Heeb[1], Miguel Cámara[1], Michael Givskov[3,4], Kim R. Hardie[1], Tim Tolker-Nielsen[4*], Bérengère Ize[2*], Paul Williams[1*]

**1** Biodiscovery Institute, National Biofilms Innovation Centre and School of Life Sciences, University of Nottingham, Nottingham, United Kingdom, **2** Laboratoire d'Ingénierie des Systèmes Macromoléculaires (LISM-UMR7255), Institut de Microbiologie de la Méditerranée, CNRS and Aix-Marseille University, Marseille, France, **3** Singapore Centre for Environmental Life Sciences Engineering, Nanyang Technological University, Singapore, **4** Costerton Biofilm Center, Department of Immunology and Microbiology, Faculty of Health Sciences, University of Copenhagen, Copenhagen, Denmark

¤ Current Address: School of Medicine, Southern University of Science and Technology, Guangdong, China
‡ These authors share first authorship on this work.
* ttn@sund.ku.dk (TT-N); bize@imsm.cnrs.fr (BI); paul.williams@nottingham.ac.uk (PW)

**Data Availability Statement:** All relevant data are within the manuscript and its Supporting Information files.

## Abstract

Extracellular DNA (eDNA) is a major constituent of the extracellular matrix of *Pseudomonas aeruginosa* biofilms and its release is regulated via pseudomonas quinolone signal (PQS) dependent quorum sensing (QS). By screening a *P. aeruginosa* transposon library to identify factors required for DNA release, mutants with insertions in the twin-arginine translocation (Tat) pathway were identified as exhibiting reduced eDNA release, and defective biofilm architecture with enhanced susceptibility to tobramycin. *P. aeruginosa tat* mutants showed substantial reductions in pyocyanin, rhamnolipid and membrane vesicle (MV) production consistent with perturbation of PQS-dependent QS as demonstrated by changes in *pqsA* expression and 2-alkyl-4-quinolone (AQ) production. Provision of exogenous PQS to the *tat* mutants did not return *pqsA*, *rhlA* or *phzA1* expression or pyocyanin production to wild type levels. However, transformation of the *tat* mutants with the AQ-independent *pqs* effector *pqsE* restored *phzA1* expression and pyocyanin production. Since mutation or inhibition of Tat prevented PQS-driven auto-induction, we sought to identify the Tat substrate(s) responsible. A *pqsA*::*lux* fusion was introduced into each of 34 validated *P. aeruginosa* Tat substrate deletion mutants. Analysis of each mutant for reduced bioluminescence revealed that the primary signalling defect was associated with the Rieske iron-sulfur subunit of the cytochrome $bc_1$ complex. In common with the parent strain, a Rieske mutant exhibited defective PQS signalling, AQ production, *rhlA* expression and eDNA release that could be restored by genetic complementation. This defect was also phenocopied by deletion of *cytB* or *cytC_1*. Thus, either lack of the Rieske sub-unit or mutation of cytochrome $bc_1$ genes results in the

**Funding:** This work was supported via grants to PW, MC and KH from the Biotechnology and Biological Sciences Research Council (BBSRC), U. K. (BB/F014392/1), BBSRC/National Biofilms Innovation Centre (BB/R012415/1), the Medical Research Council U.K. (MR/N501852/1), the Wellcome Trust (103884/Z/14/Z and 108876/Z/15/Z) and the European Union FP7 collaborative action grant (NABATIVI, 223670). BI was supported by a grant from Vaincre la Mucoviscidose and the Grégory Lemarchal associations (RF20140501138). TTN and MG were supported by grants from the Danish Strategic Research Council (9040-00023B), the Danish Council for Independent Research (09-065732), the Novo Nordisk Foundation (Biofilm2-AEEE/CPH) and the Lundbeck Foundation (2015-486), Denmark. The funders had no role in study design, data collection and analysis, decision to publish, or preparation of the manuscript.

**Competing interests:** No authors have competing interests

perturbation of PQS-dependent autoinduction resulting in eDNA deficient biofilms, reduced antibiotic tolerance and compromised virulence factor production.

## Author summary

*Pseudomonas aeruginosa* is a highly adaptable human pathogen responsible for causing chronic biofilm-associated infections. Biofilms are highly refractory to host defences and antibiotics and thus difficult to eradicate. The biofilm extracellular matrix incorporates extracellular DNA (eDNA). This stabilizes biofilm architecture and helps confer tolerance to antibiotics. Since mechanisms that control eDNA release are not well understood, we screened a *P. aeruginosa* mutant bank for strains with defects in eDNA release and discovered a role for the twin-arginine translocation (Tat) pathway that exports folded proteins across the cytoplasmic membrane. Perturbation of the Tat pathway resulted in defective biofilms susceptible to antibiotic treatment as a consequence of perturbed pseudomonas quinolone (PQS) signalling. This resulted in the failure to produce or release biofilm components including eDNA, phenazines and rhamnolipids as well as microvesicles. Furthermore, we discovered that perturbation of PQS signalling was a consequence of the inability of *tat* mutants to translocate the Rieske subunit of the cytochrome $bc_1$ complex involved in electron transfer and energy transduction. Given the importance of PQS signalling and the Tat system to virulence and biofilm maturation in *P. aeruginosa*, our findings underline the potential of the Tat system as a drug target for novel antimicrobial agents.

## Introduction

*Pseudomonas aeruginosa* is an opportunistic pathogen that causes a wide range of human infections including lung, urinary tract and wound, bacteremia and infections associated with medical devices [1]. It is notorious for its tolerance to antimicrobial agents, a property that is largely a consequence of its ability to form biofilm communities [1,2]. Bacterial exoproducts including cell surface appendages, extracellular polymeric substances, biosurfactants and secondary metabolites all contribute to *P. aeruginosa* biofilm formation and maturation [3–7].

Apart from exopolysaccharides such as Psl, Pel and alginate, the extracellular polymeric matrix of *P. aeruginosa* biofilms incorporates proteins, rhamnolipids, membrane vesicles (MVs) and extracellular DNA (eDNA) [5,8–10]. Rhamnolipid biosurfactants are required during the initial stages of micro-colony formation and are involved in the migration-dependent formation of the caps of mushroom-shaped micro-colonies formed in flow-cell grown biofilms [10]. They also aid the maintenance of channels between multicellular structures within biofilms and contribute to biofilm dispersal [5]. With respect to the biofilm micro-colonies that characteristically form in flow-chambers fed with glucose minimal medium, eDNA is present at high concentrations in the outer layers of microcolonies in young biofilms. However, in mature biofilms, eDNA is primarily located in the stalks at the borders between micro-colony caps and stalks [8].

The release of eDNA occurs via the lysis of a sub-population of bacterial cells [10–14]. It is involved in attachment, aggregation and stabilization of biofilm microcolonies. eDNA can act as a nutrient source, chelate metal cations and confer tolerance to antibiotics such as the polymyxins and aminoglycosides [10,12,13]. eDNA also binds other biopolymers

(exopolysaccharides and proteins) stabilizing biofilm architecture and conferring protection against adverse chemical and physical challenges [12,13]. By intercalating with eDNA, secondary metabolites such as phenazines enhance biofilm integrity [12,15]. Pyocyanin for example can contribute to DNA release through the formation of reactive oxygen species such as hydrogen peroxide that damage cell membranes [12]. Although the mechanism(s) responsible for eDNA release has not been fully elucidated both eDNA and MVs can be generated via explosive cell lysis mediated via a cryptic prophage endolysin encoded within the R- and F-pyocin gene clusters [14].

In *P. aeruginosa*, rhamnolipids and pyocyanin production, eDNA and MV release, and hence biofilm development, are all controlled by quorum sensing (QS) [1,8,16]. Consequently, *P. aeruginosa* mutants with defects in this cell-to-cell communication system form aberrant, flat undifferentiated biofilms [10]. In *P. aeruginosa*, the QS regulatory network consists of a hierarchical cascade incorporating the overlapping Las, Rhl and PQS pathways that employ *N*-acylhomoserine lactones (AHLs) and 2-alkyl-4-quinolones (AQs) as signal molecules [1,16,17]. All three QS systems contain auto-induction loops whereby activation of a dedicated transcriptional regulator by the cognate QS signal molecule induces expression of the target synthase such that QS signal molecule production can be rapidly amplified to promote co-ordination of gene expression at the population level.

*P. aeruginosa* produces a diverse family of AQs and AQ *N*-oxides [18] of which 2-heptyl-3-hydroxy-4-quinolone (the *Pseudomonas* Quinolone Signal, PQS) and its immediate precursor 2-heptyl-4-hydroxyquinoline (HHQ) are most closely associated with PQS signalling [17] (**Fig 1**). Most of the genes required for AQ biosynthesis (*pqsABCDE*) and response (*pqsR/mvfR*) are located at the same genetic locus although *pqsH* and *pqsL* are distally located [17]. The biochemical basis for AQ and AQ *N*-oxide biosynthesis is summarized in **S1 Fig**. PqsA catalyses the formation of anthraniloyl-CoA that is condensed with malonyl-CoA by PqsD to form 2-aminobenzoylacetyl-CoA (2-ABA-CoA) [19,20]. The latter is converted to 2-amino-benzoylacetate (2-ABA) via the thioesterase functionality of PqsE [21]. The PqsBC heterodimer condenses 2-ABA with octanoyl-CoA to generate HHQ [22,23]. PQS is formed through the oxidation of HHQ by PqsH [24] while formation of the AQ *N*-oxides such as 2-heptyl-4-hydroxyquinoline *N*-oxide (HQNO) requires the alternative mono-oxygenase PqsL [25]. The PqsE protein has dual functions; while it is not essential for AQ biosynthesis, it is required for the AQ-independent production of several factors that contribute to biofilm maturation including pyocyanin, rhamnolipids and lectin A [26]. While activation of RhlR-dependent genes depends on PqsE [27], the AQ-independent, thioesterase-independent mechanism by which PqsE acts has not yet been elucidated [17,21].

The *pqs* system is subject to positive autoinduction, since the LysR-type transcriptional regulator PqsR (MvfR), binds to the promoter region of *pqsABCDE* (P*pqsA*) triggering transcription once activated by HHQ or PQS [28–30] (**Fig 1**). Therefore, by analogy with other QS systems, HHQ and PQS can both act as autoinducers by generating a positive feedback loop that accelerates their biosynthesis and co-ordinates a population-wide response. However, in contrast to HHQ which only regulates the *pqsABCDE* operon [17], PQS is a ferric iron chelator [31] that not only drives AQ biosynthesis via PqsR but also the expression of genes involved in the iron-starvation response and virulence factor production *via* PqsR-dependent and PqsR-independent pathways [17]. In addition, PQS can act as a cell-sensitizing pro-oxidant [32] and is required for MV production via a direct physical interaction with lipopolysaccharide (LPS) within the outer membrane [33]. The packaging of PQS within MVs also provides a means for trafficking this hydrophobic QS signal within a *P. aeruginosa* population [34].

With respect to biofilm development, PQS signalling is of particular interest because *pqsA* biosynthetic mutants fail to produce eDNA, rhamnolipids, pyocyanin and MVs, and form thin

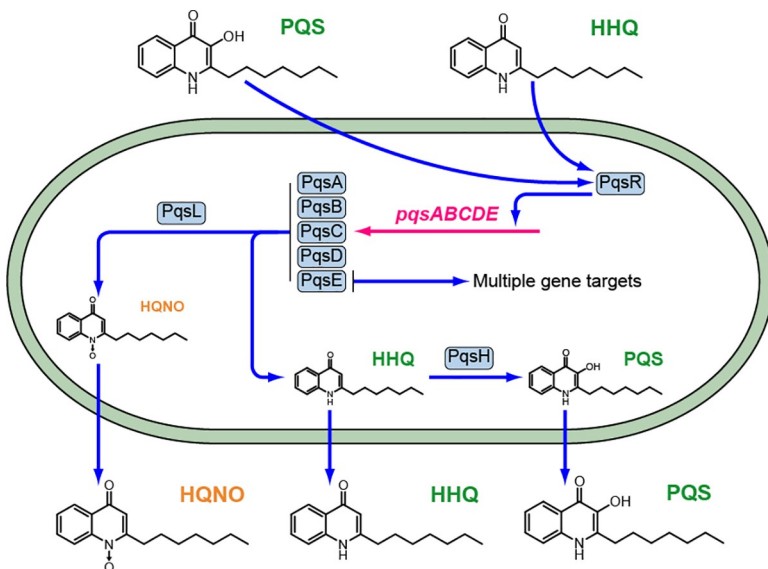

**Fig 1. The PQS signalling pathway in *P. aeruginosa*.** The PqsABCDE proteins synthesize HHQ, which is converted to PQS by PqsH and also HQNO in conjunction with PqsL. Both HHQ and PQS are released by the cells into the extracellular environment and are taken back up by neighboring cells. Autoinduction occurs when either HHQ or PQS binds to PqsR and amplifies expression of the *pqsABCDE* operon. The *pqsE* gene product has dual functions contributing to AQ biosynthesis as a thioesterase and also via an AQ-independent, thioesterase-independent mechanism for e.g. pyocyanin, rhamnolipid and lectin production as well as biofilm maturation. The conversion of HHQ to PQS confers additional functionalities since PQS unlike HHQ induces microvesicle formation and is a potent iron chelator.

defective biofilms containing little eDNA [8, 27,33,35]. The mechanism involved in PQS-mediated DNA-release in biofilms is not understood but has been suggested to be linked to phage induction causing cell lysis [8,36–40]. Although explosive cell lysis releases eDNA in biofilms and generates MVs through vesicularization of shattered membrane fragments, *pqsA* mutants are not defective for explosive lysis [14] and therefore this phenomenon is unlikely to account for PQS-dependent eDNA release.

In the present study we sought to identify additional factors involved in eDNA release by screening a transposon (Tn) mutant library for eDNA-release defective mutants. Apart from *pqs* biosynthetic mutants, we obtained Tn insertion mutants within the twin-arginine translocation (Tat) pathway that exhibited reduced levels of eDNA release, fail to produce rhamnolipids or pyocyanin and form defective, eDNA- poor, antibiotic susceptible biofilms. Since mutation or deletion of *tat* resulted in altered AQ production, reduced pyocyanin, rhamnolipid and MVs, and as the *tat* mutants were refractory to exogenously supplied PQS, the aberrant biofilm phenotype observed could be accounted for by perturbation of PQS autoinduction. By screening a library of *P. aeruginosa* Tat substrate mutants, we identified the Rieske sub-unit of the cytochrome $bc_1$ complex as the Tat substrate required for PQS-dependent QS and hence eDNA release and biofilm maturation.

## Results

### Transposon mutagenesis screen for *P. aeruginosa* mutants exhibiting reduced DNA release

To identify *P. aeruginosa* genes that contribute to eDNA release, a mariner Tn mutant library was generated in strain PAO1. Approximately 10,000 mutants grown in microtitre plates were

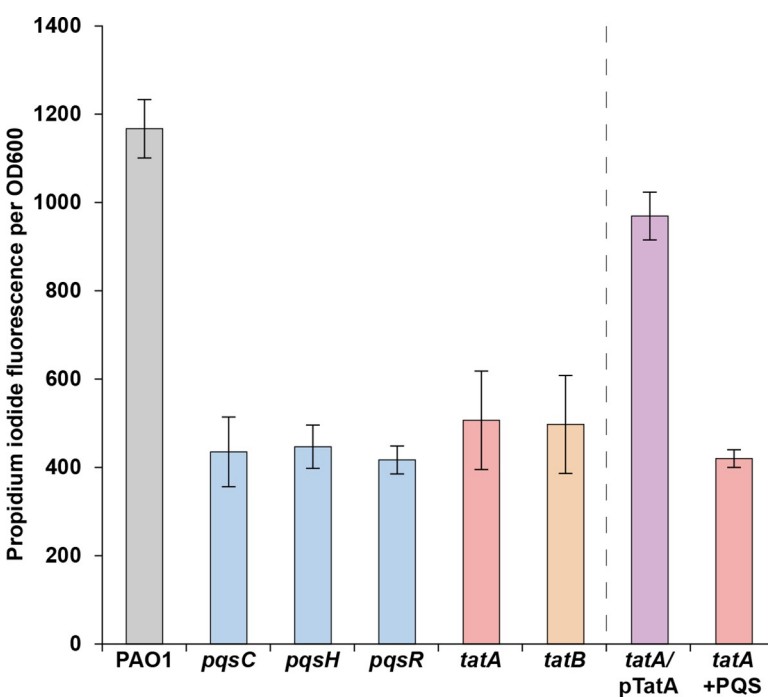

**Fig 2. Transposon mutant screen for *P. aeruginosa* strains defective for eDNA release.** *P. aeruginosa* wild-type and mutant strains were grown for 24 h in 96 well microtiter plates containing ABTG medium, after which the relative levels of eDNA in the cultures were determined using a PI binding assay. The means and standard deviations of eight replicates are shown.

assayed for reduced eDNA release using propidium iodide (PI) to quantify eDNA because it is unable to penetrate live bacteria and its fluorescence is enhanced 30-fold on binding DNA [41]. From the initial screen, 84 Tn insertion mutants were selected and re-screened to eliminate strains with double Tn insertions and to confirm their eDNA phenotype. For each of the remaining 34 mutants exhibiting reduced eDNA, the regions flanking each Tn insertion were sequenced and the corresponding genes identified. For most of the eDNA-release deficient mutants, the Tn insertions were located within genes required for AQ biosynthesis (*pqsC* and *pqsH)* or regulation (*pqsR*) (**Fig 2**) These data confirm our previous work that first uncovered a role for PQS signalling in eDNA release [8].

Apart from the *pqs* mutants, two mutants were obtained with insertions in the *tatA* and *tatB* genes respectively (**Fig 2**) that code for components of the twin-arginine translocation (Tat) system. Tat exports folded proteins out of the cytoplasm and across the cytoplasmic membrane in an ATP-independent manner [42]. It was originally named with respect to the presence of an Arg-Arg motif in the signal sequence of Tat-exported products (sometimes called Tat substrates) [42,43]. In *P. aeruginosa* the Tat translocase complex consists of three proteins (TatA, B and C) [43]. TatB and TatC form the receptor complex for Tat substrate precursors whereas TatA functions as the main facilitator for protein translocation across the membrane [42]. In *P. aeruginosa* diverse proteins involved in phosphate and iron metabolism, virulence and energy transduction are exported to the periplasm, or secreted via the Tat export system and *tat* mutants exhibit pleiotropic phenotypes [44,45]. **Fig 2** shows that genetic complementation of the *P. aeruginosa tatA* mutant with a plasmid-borne copy restored eDNA release.

## The Tat pathway contributes to biofilm development and tobramycin susceptibility in *P. aeruginosa*

Since eDNA makes an important contribution to biofilm development and architecture [10,12,13], biofilm formation by the *P. aeruginosa tatA* mutant in flow-chambers was investigated. After 4-days growth, the *P. aeruginosa* wild-type and complemented *tatA*/pTatA mutant formed biofilms with mushroom-shaped structures whereas the *tatA* mutant formed thin, flat biofilms (**Fig 3A**). In addition, eDNA was observed primarily in the stalks of the mushroom-shaped structures in the wild-type whereas *tatA* mutant biofilms contained no stalks and little extracellular DNA. Consistent with this flow-cell biofilm phenotype, exposure of the biofilms formed by each of the three strains to tobramycin showed that the *tatA* mutant biofilm was more sensitive to tobramycin than either the wild-type or *tatA*/pTatA mutant biofilm (**Fig 3B**). Since eDNA binds positively charged antibiotics and as exogenously provided DNA increases aminoglycoside tolerance by integrating into *P. aeruginosa* biofilms, the increased sensitivity to tobramycin is likely to be a consequence of the reduction in eDNA within the biofilm extracellular matrix [46,47].

## *P. aeruginosa tatA* mutants are defective in the production of rhamnolipids, pyocyanin and MVs

Since rhamnolipids, pyocyanin and MVs are all important components of *P. aeruginosa* biofilms, their production was quantified in the *tatA* Tn insertion mutant and in a Δ*tatABC*

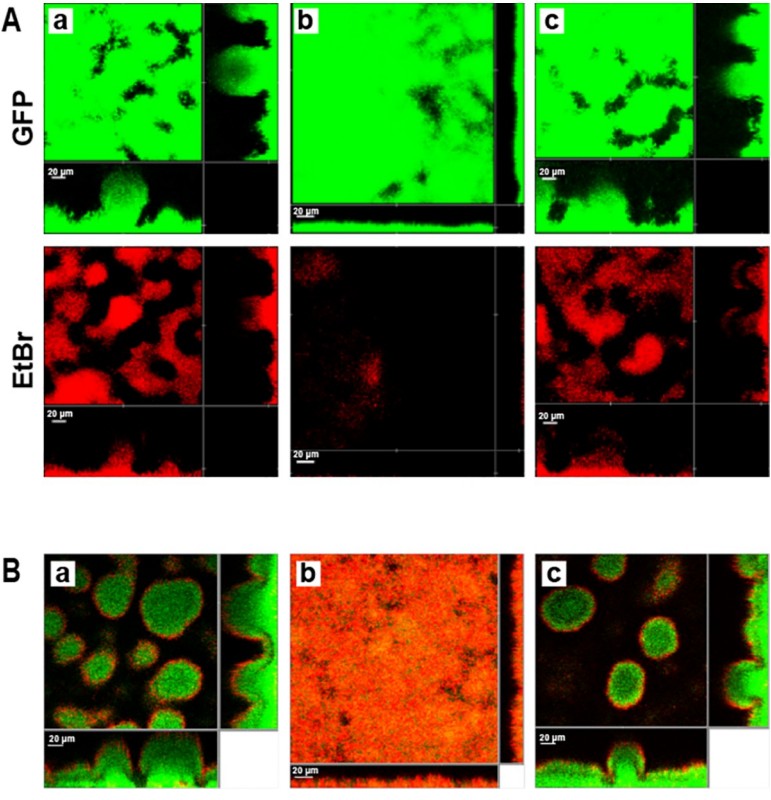

**Fig 3. *P. aeruginosa tat* mutants form defective biofilms with increased susceptibility to tobramycin.** CLSM images showing four-day-old biofilms formed in flow chambers of the *gfp*-tagged *P. aeruginosa* wild-type (a), *tatA* mutant (b) and genetically complemented *tatA* mutant (c). In **(A)** biofilms were stained for total biomass with Syto9 (green) and for eDNA with ethidium bromide (red). In **(B)** biofilms were treated with tobramycin and the medium was supplemented with propidium iodide prior to CLSM such that dead cells appear red while live cells appear green. Each panel shows one horizontal optical section two flanking vertical optical sections. Bars, 20 μm.

deletion mutant. **Fig 4A and 4B** show that *tat* mutants produce substantially less pyocyanin and rhamnolipid than the parent or *tatA* complemented strain. Furthermore, MV levels (**Fig 4C**) were reduced by ~50% in the *tatA* mutant compared with the wild type and could be restored by genetic complementation.

## Inactivation of the Tat pathway by mutation or small molecule-mediated inhibition perturbs PQS signalling

The reductions in eDNA release, rhamnolipids, pyocyanin and MVs noted in the *tat* mutant as well as its biofilm phenotype are comparable with those observed for *P. aeruginosa* strains with mutations in *pqs* genes such as *pqsA*, the first gene in the AQ biosynthetic pathway (see **Figs 1** and **4** and [8, 40]). These data therefore suggested that the *tat* mutant biofilm phenotype was likely, at least in part, to be due to a defect in PQS signalling. To investigate the impact of the *tatA* mutation on the expression of *pqsA*, a CTX::*pqsA'-lux* fusion was introduced into the chromosomal CTX site of both the wild type and *tatA* mutant. **Fig 5A** shows that *pqsA* expression in the *tatA* mutant is reduced ~4 fold compared with the wild type strain and restored by genetic complementation of the mutant. In agreement with these data, the Tat inhibitor, Bayer 11–7082, identified by Vasil *et al* [48] reduced *pqsA* expression in the wild type PAO1 strain by ~4 fold consistent with the reduction noted for the CTX::*pqsA'-lux* fusion in the *tatA* mutant (**Fig 5B**). Bayer-11 7082 had no effect on growth or light output in *P. aeruginosa* expressing the *lux* genes from a derepressed *lac* promoter (**S2 Fig**). In addition, the concentration of PQS in whole culture extracts of *P. aeruginosa* after growth in LB medium as determined by LC-MS/MS was respectively ~56% lower in the *tatA* mutant compared with the wild type and complemented *tat* mutant (**Fig 5C**). Since *pqsA* expression and hence AQ production is also PqsR/MvfR-dependent, we compared the expression of *pqsR* in the Δ*tatABC* with the parent strain but found no difference (**S3 Fig**).

## Exogenous PQS does not restore PQS signalling in a *P. aeruginosa tatA* mutant

QS systems are characteristically autoinducible such that exogenous provision of the cognate signal molecule usually induces expression of the signal synthase and hence activation of downstream target genes [49]. When the *tatA* mutant was provided with exogenous PQS, eDNA release did not increase (**Fig 2**). To investigate this finding further, either PQS or HHQ was exogenously supplied to the wild type, *tatA* mutant or the complemented *tatA* mutant strains carrying chromosomal *pqsA'-lux* fusions. The data presented in **Fig 6A** show that the response of the *tatA* mutant to PQS or HHQ respectively at 5 or 20 μM with respect to *pqsA* expression was at least 2-fold lower than the controls. Since both wild type and *tatA* mutant still produce AQs endogenously, the experiments were repeated in the *P. aeruginosa* Δ*pqsA* and *tatA* Δ*pqsA* mutants since no AQs are produced in these genetic backgrounds. **Fig 6B** shows that the response to both PQS and HHQ is substantially reduced (e.g. ~3 fold at 5 μM PQS) in the absence of *tatA* in the *P. aeruginosa tatA* Δ*pqsA* double mutant. This reduced response to PQS could be due either to reduced uptake or the inability to respond to the exogenous QS signal molecule.

To determine the consequences of perturbed PQS signalling on the expression of the rhamnolipid (*rhlA*) and pyocyanin biosynthetic genes (*P. aeruginosa* has two, almost identical redundant 7 gene phenazine biosynthetic operons termed *phzA1-G1* and *phzA2-G2;* [50]), the corresponding miniCTX-*lux* promoter fusions for *rhlA* and *phzA1* respectively were constructed and introduced onto the chromosomes of the wild type, Δ*pqsA* and *tatA* Δ*pqsA* mutants respectively. **Fig 7A** shows the expression profiles of *rhlA'-lux* as a function of time.

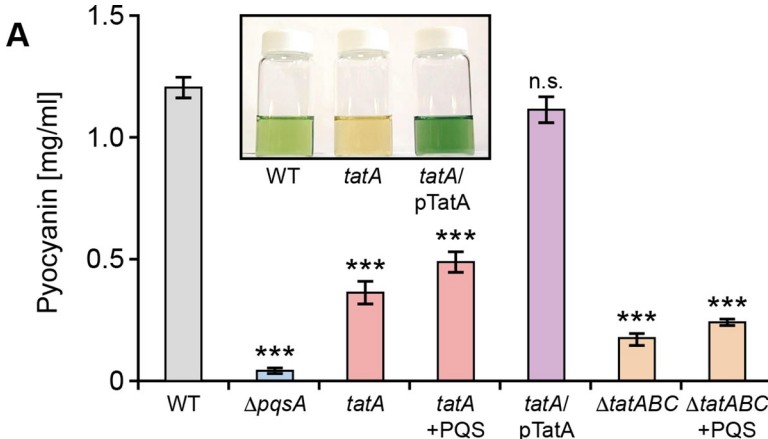

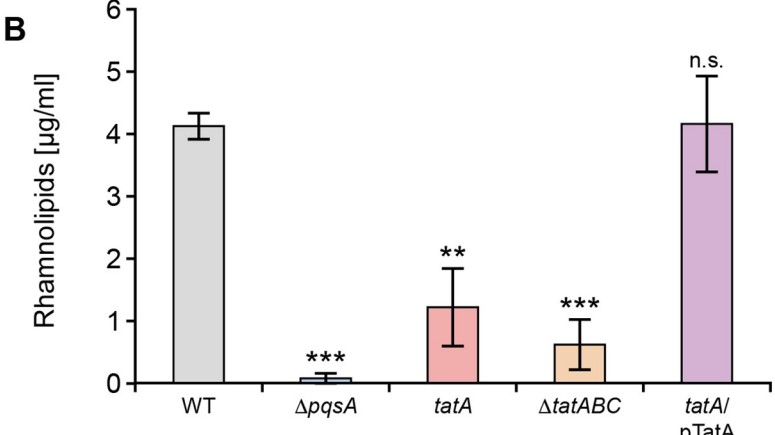

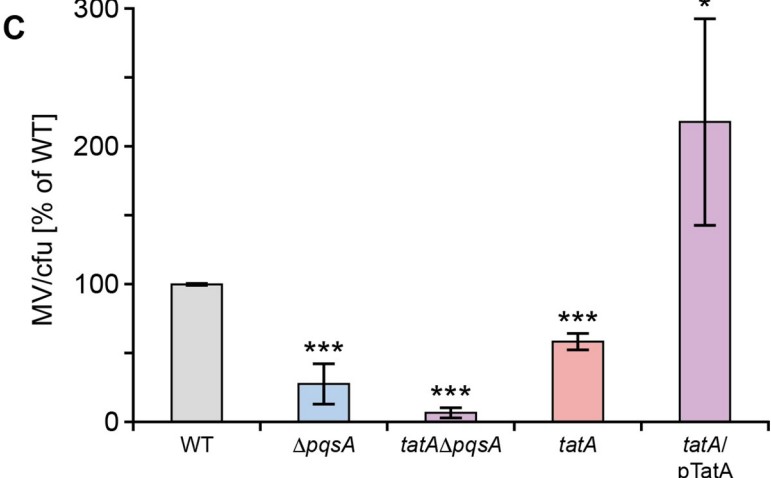

**Fig 4.** Production of pyocyanin (**A**), rhamnolipids (**B**) and MVs (**C**) are reduced in *P. aeruginosa tat* mutants. (**A**) Pyocyanin levels are shown in the *P. aeruginosa* wild type, Δ*pqsA*, *tatA*, and *tatABC* deletion mutants and the *tatA* mutant complemented with plasmid-borne *tatA*. The impact of exogenous PQS (40 μM) on the *tatA* and Δ*tatABC* mutants is also shown. Insert panel shows the absence of green pigment in the *tatA* mutant compared with the wild type and complemented *tatA* mutant. (**B**) Rhamnolipid production in the *tatA* and Δ*tatABC* mutants compared with

the wild type, Δ*pqsA* mutant and *tatA* complemented with plasmid-borne *tatA*. (**C**) Comparison of MV production in the *tatA* mutant, complemented *tatA* mutant and in a double *tatA* Δ*pqsA* mutant compared with the wild type strain. Experiments were repeated in triplicate at least twice. ***p < 0.001, **p < 0.01; n.s. not significant.

Both the wild type and Δ*pqsA* mutant show an ~2 fold increase in *rhlA* expression when supplied with exogenous PQS (20 μM) and share similar profiles over the growth curve. In contrast, the *rhlA'-lux* fusion in the *tatA* mutant does not show the same expression profile or response to exogenous PQS as the wild type and Δ*pqsA* mutant strains. The *rhlA'-lux* expression profile in the *tatA* mutant supplied with exogenous PQS is however clearly restored when the mutation is complemented by pTatA (**Fig 7A**).

AQ-dependent QS is required for *phzA1* expression [27,50]. Exogenous PQS increased *phzA1'-lux* expression by ~4 fold in both wild type and Δ*pqsA* mutant backgrounds (**Fig 7B**). However, the *tatA* Δ*pqsA* double mutant responded comparatively poorly to PQS (**Fig 7B**).

## Constitutive expression of *pqsABCD* does not restore AQ biosynthesis in a *tat* mutant

Since mutation of *tatA* resulted in reduced *pqsA* expression, it was possible that the auto-induction of PQS biosynthesis via PqsR (MvfR) is compromised. To uncouple the autoinduction of AQ production, the *pqsABCD* genes were introduced into the *P. aeruginosa* Δ*pqsA* and Δ*pqsA*Δ*tatA* mutants respectively on a plasmid (pBBRMCS5::*pqsABCD*) constitutively expressing *pqsABCD* [51]. **Fig 8** shows that PQS, HHQ and HQNO are present in the culture medium of the Δ*pqsA* mutant transformed with pBBRMCS5::*pqsABCD*. However neither the cell free supernatant (**Fig 8**) nor whole cells of the *tatA* Δ*pqsA* double mutant transformed with pBBRMCS5::*pqsABCD* contained or accumulated intracellular AQs (**S4 Fig**). These data suggested that in a *tat* mutant background, the lack of AQs is not a consequence of an AQ transport defect, but is due to the inability to fully activate AQ biosynthesis at the appropriate time/population density. In the absence of an auto-inducible *pqs* system in a *tat* mutant background, the defect in AQ biosynthesis and hence PQS signalling appears to be more severe.

## Reduced rhamnolipid production does not account for defective PQS signalling

In *P. aeruginosa* biofilms, rhamnolipids provide protective shielding against neutrophils [52,53] and contribute to the effectiveness of PQS signalling by increasing the solubility and bioactivity of PQS [54]. In **Fig 4B**, we showed that rhamnolipid production was substantially reduced in the *P. aeruginosa tat* mutant background. To determine whether the perturbation of PQS signalling in the *tat* mutants was a consequence of reduced rhamnolipid production, we investigated the impact of exogenous rhamnolipids on *pqsA* expression. **S5 Fig** shows that the addition of purified rhamnolipids (10 or 50 μg/ml) to the *tatA* Δ*pqsA* mutant with or without PQS (40 μM) had little effect on PQS signalling indicating that the defect in the *tat* mutants was not simply due to the loss of rhamnolipid production and an inability to solubilize PQS.

## PqsE restores pyocyanin in the *tat* mutants

Although PqsE is not essential for AQ biosynthesis, it is an effector protein required for the production of pyocyanin, rhamnolipids and biofilm maturation and its function is independent of PQS, HHQ or PqsR [27,55]. Consequently, the expression of PQS -dependent exoproducts such as pyocyanin can be restored in a *pqsA* negative (and hence AQ-negative) mutant by expressing a plasmid-borne copy of *pqsE*. In the *tatA* Δ*pqsA* double mutant, *phzA1*

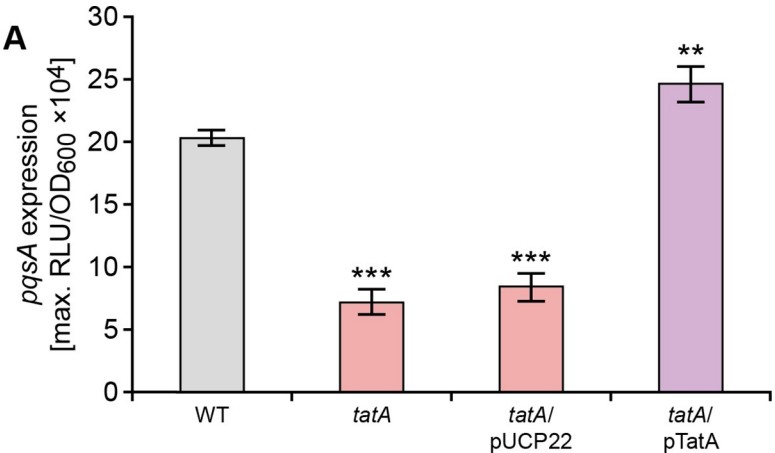

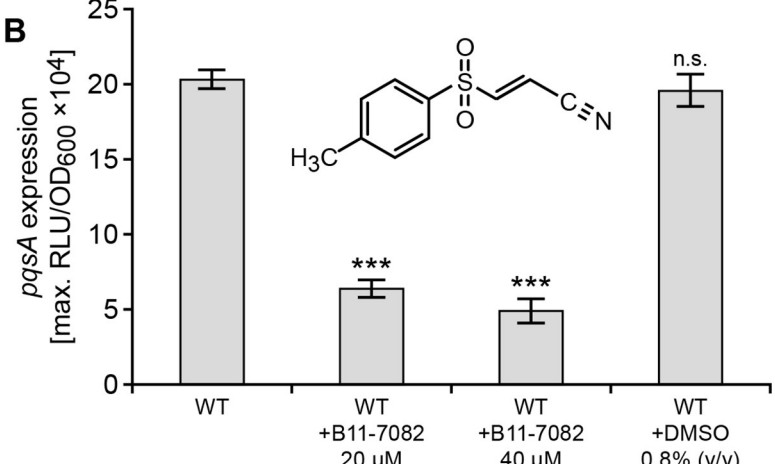

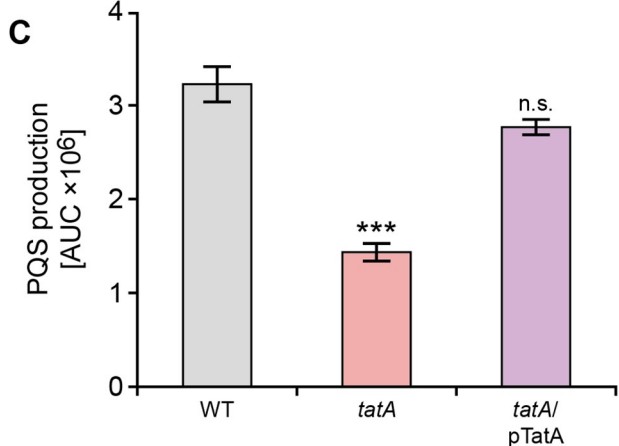

**Fig 5. Mutation of *tatA* or exposure to the Tat inhibitor Bayer 11–7082 inhibits *pqsA* expression and AQ production.** (**A**) Mutation of *tatA* or (**B**) treatment with Bayer 11–7082 supplied at either 20 μM or 40 μM reduces the maximal expression of a *P. aeruginosa* PAO1 chromosomal *pqsA'-lux* promoter fusion without affecting growth. (**C**) LC-MS/MS analysis of PQS production by *P. aeruginosa* PAO1 wild type compared with the *tatA* mutant and complemented *tatA* mutant. Experiments were repeated in triplicate at least twice. ***p < 0.001; n.s. not significant.

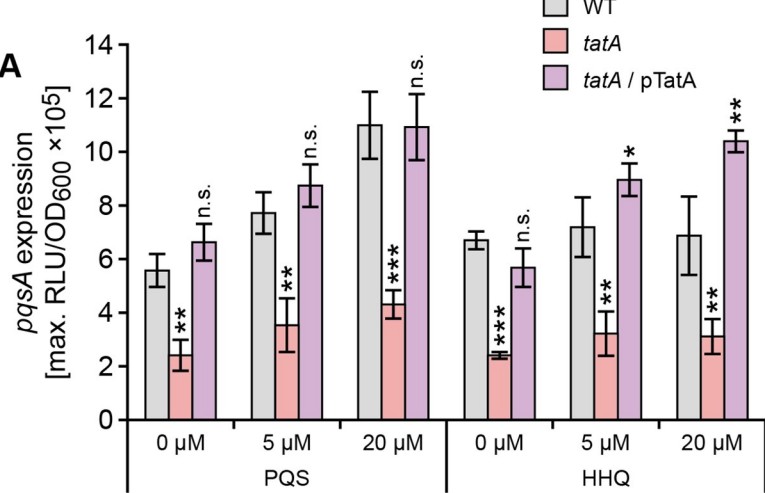

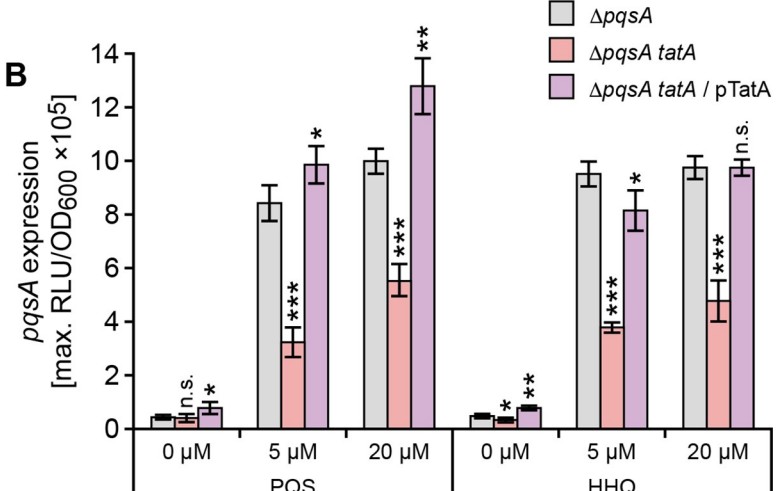

**Fig 6.** Exogenous AQs do not restore *pqsA* expression in *P. aeruginosa tatA* (A) or *tatA* Δ*pqsA* (B) mutants. Exogenous PQS or HHQ was added at 5 μM or 20 μM to (**A**) wild type, the *tatA* mutant and complemented *tatA* mutant or (**B**) Δ*pqsA*, Δ*pqsA tatA* or *tatA* Δ*pqsA* mutant complemented with *tatA*. Maximal light output from the chromosomal *pqsA'-lux* fusion was recorded as a function of growth (RLU/OD$_{600}$). Experiments were repeated in triplicate at least twice. ***p < 0.001, **p < 0.01, and *p < 0.05; n.s. not significant.

expression (**Fig 7B**) and pyocyanin production (**Fig 9**) were respectively restored by *pqsE* expression indicating that the *tat* mutation does not affect PqsE functionality.

## Identification of the Tat substrate responsible for perturbation of PQS signalling

Recently Gimenez et al [45] experimentally validated the Tat-mediated export of 34 *P. aeruginosa* gene products predicted to have Tat signal peptides. To determine which of the exported Tat substrates was responsible for perturbation of PQS signalling, allelic replacement mutants were constructed in *P. aeruginosa* strain PA14 for each substrate. Before introducing the

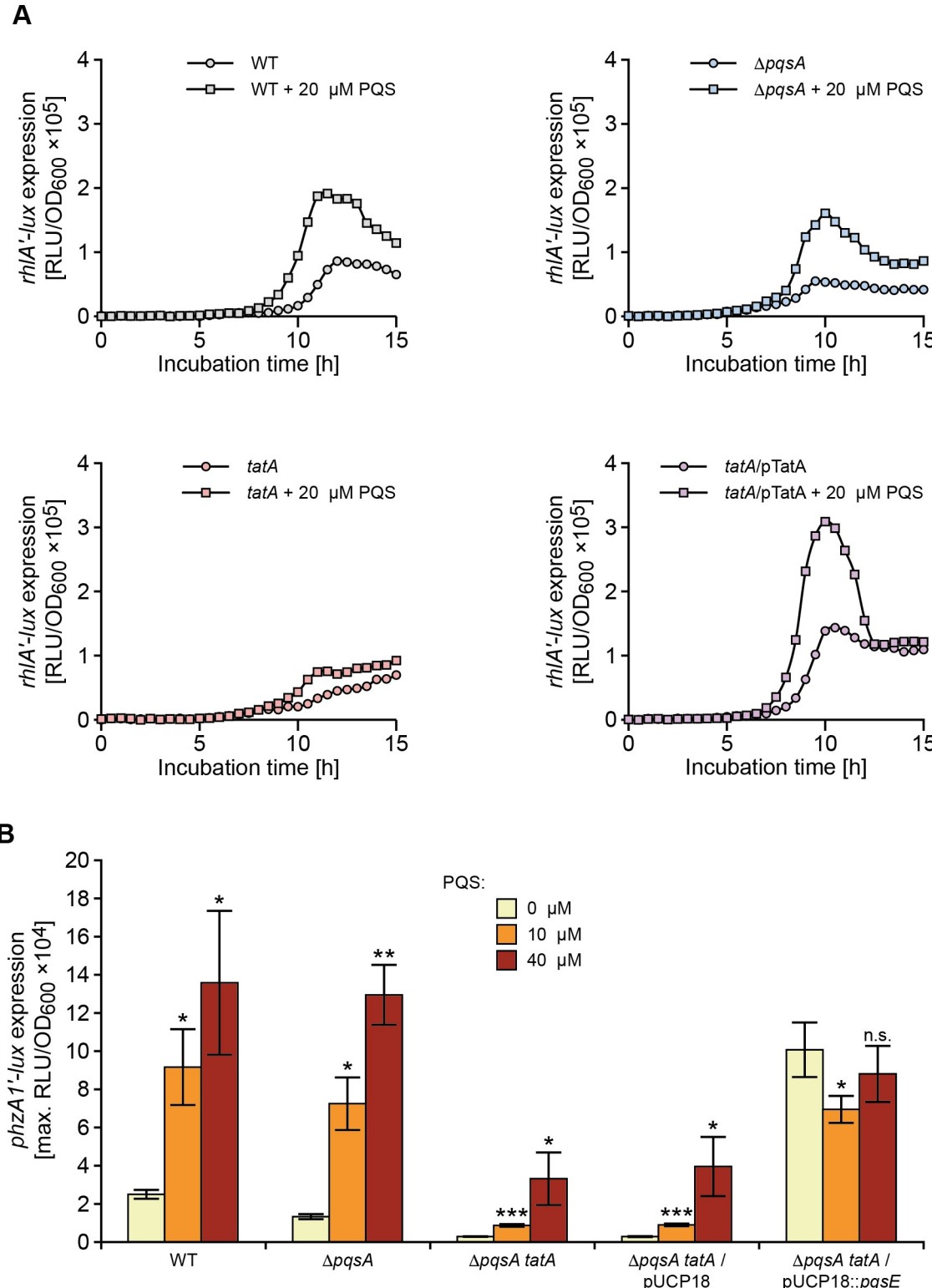

**Fig 7. Rhamnolipid (*rhlA*) and pyocyanin biosynthetic (*phzA1*) genes show altered expression profiles in *P. aeruginosa tat* mutants and fail to respond to exogenous PQS.** (**A**) Light output from a chromosomal *rhlA'-lux* fusion as a function of growth (RLU/OD) over time when introduced into the wild type, *pqsA* and *tatA* mutants or complemented *tatA* mutant in the absence or presence of exogenous PQS (20 μM). (**B**) Maximal light output from a chromosomal *phzA1'-lux* fusion as a function of growth (RLU/OD$_{600}$) when introduced into the wild type, Δ*pqsA*, or Δ*pqsA tatA* mutants in the absence or presence of exogenous PQS (10

or 40 μM) or plasmid-borne *pqsE* or the pUCP18 vector control. Experiments were repeated in triplicate at least twice. ***p < 0.001, **p < 0.01, and *p < 0.05; n.s. not significant.

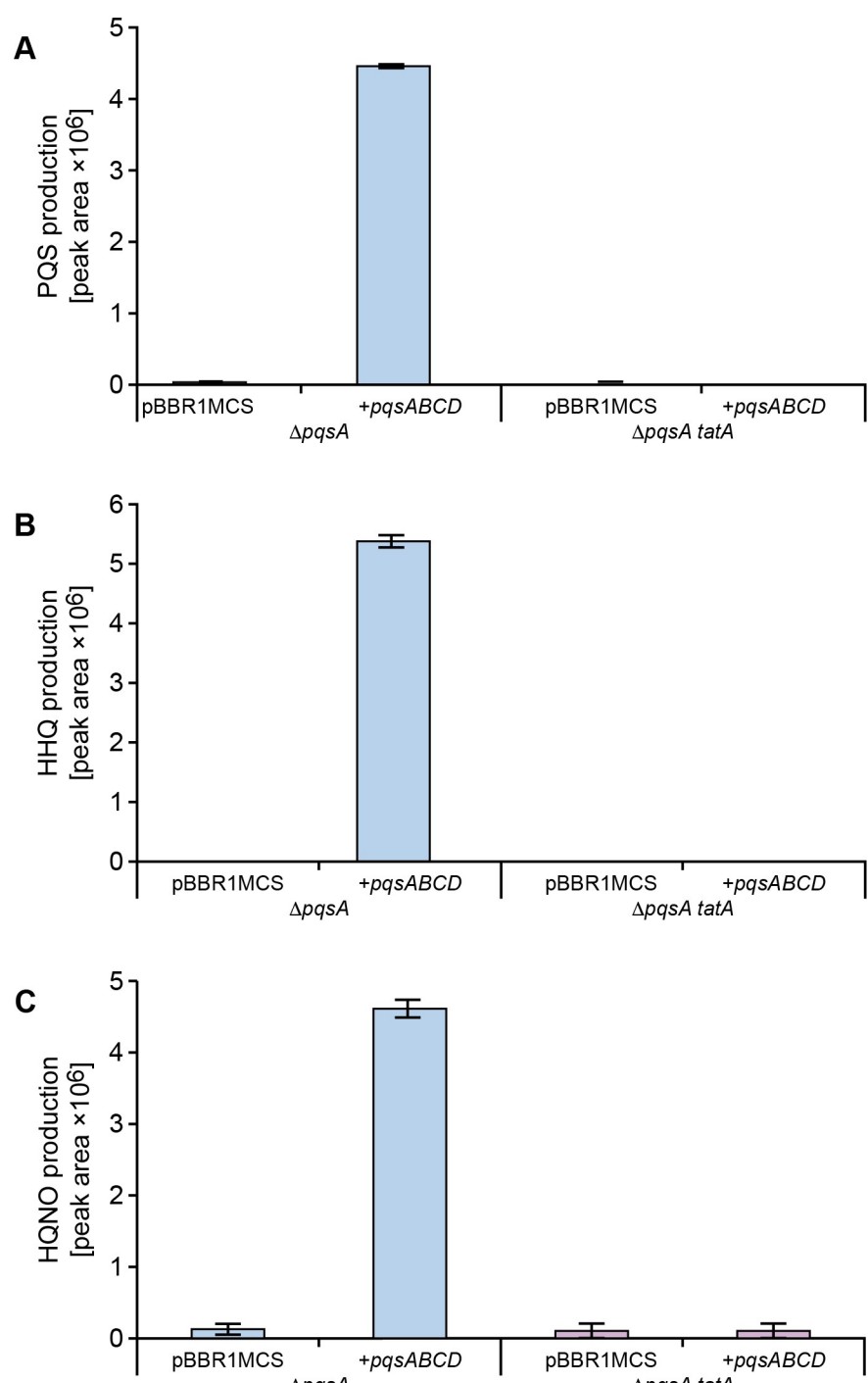

**Fig 8. AQ biosynthesis is not restored in a *tatA ΔpqsA* double mutant by plasmid-borne *pqsABCD* in the absence of autoinduction.** Semi-quantitative analysis by LC-MS/MS of PQS, HHQ and HQNO production by *P. aeruginosa pqsA* and *tatA ΔpqsA* mutants respectively without or with the *pqsABCD* biosynthetic genes provided *in trans* via pBBR1MCS-5::*pqsABCD*. Experiments were repeated in triplicate at least twice.

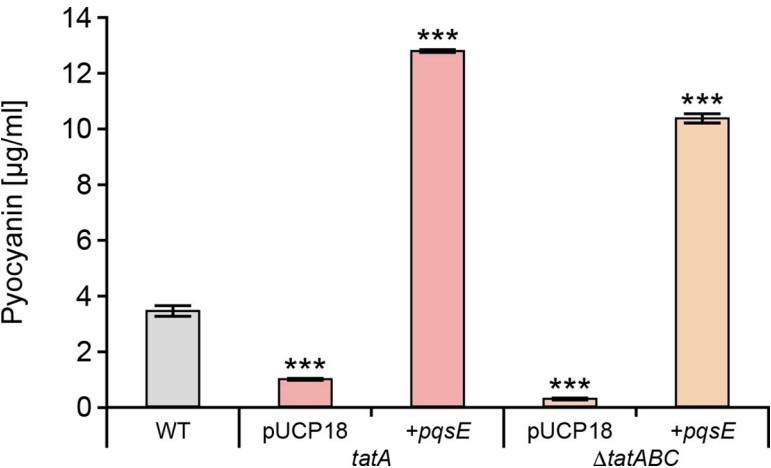

**Fig 9. PqsE restores pyocyanin production in *P. aeruginosa tat* mutants.** Production of pyocyanin by the wild type compared with the *tatA* and Δ*tatABC* mutants transformed with plasmid-borne *pqsE* or the empty vector. Experiments were repeated in triplicate at least twice. ***p < 0.001.

*pqsA'-lux* fusion onto the chromosomal CTX site of each Tat substrate mutant, we first confirmed that PQS signalling in a PA14 Δ*tatABC* mutant was perturbed in a similar manner to that observed for PAO1, the genetic background used so far in the study (**Fig 10A**). Determination of the maximum expression of *pqsA'-lux* for each of the 34 Tat substrate mutants (**Fig 10A**) revealed that although a number of mutants exhibited reduced light output, the greatest reduction was observed for deletion of PA14_57570 (equivalent to PA4431 in PAO1, here designated *petA* following the nomenclature of orthologues described in other species). This gene codes for the Rieske subunit of the cytochrome $bc_1$ complex that is involved in electron transfer and respiration under aerobic conditions and is also required for *P. aeruginosa* growth under anaerobic conditions in the presence of nitrite, nitric oxide or nitrous oxide [56].

To confirm that the Δ*petA* mutant in common with the *P. aeruginosa tat* mutants exhibited similar defects as a consequence of perturbed PQS signalling, we compared their AQ production (**Fig 10B**), *rhlA* expression, and eDNA release in biofilms (**S6** and **S7** **Figs**). Comparison of PQS, HHQ and HQNO production in PA14 with the Δ*petA* mutant and the strain with a *cis* complementation of the *petA* mutation at the CTX site confirmed the loss and increase in AQ production (**Fig 10B**). Similar results were obtained for *pqsA* expression in a PAO1 Δ*petA* mutant (**S8 Fig**). In *P. aeruginosa* PA14, the *rhlA* expression profiles for the parent and Δ*tatABC* deletion mutant in the absence or presence of exogenously supplied PQS were similar to those of strain PAO1 (compare **S6 Fig** with **Fig 7A**). Confocal microscopy images of biofilm formation by *P. aeruginosa* PA14 under static growth conditions shows that in common with the Δ*tatABC* mutant, Δ*petA* mutant biofilms lack the eDNA content observed in the wild type and complemented mutant (**S7 Fig**) demonstrating that PetA is the primary Tat substrate required for PQS-dependent eDNA release. Since PetA is one of the three electron transfer proteins that constitute the cytochrome $bc_1$ complex we also investigated whether mutation of *cytB* and *cytC₁* genes (PA4429 and PA4430 respectively) perturbed PQS-signalling. **S9 Fig** shows Δ*cytB* and Δ*cytC*₁ mutants also exhibited reduced *pqsA* expression and AQ production indicating that a fully functional cytochrome $bc_1$ is essential for PQS-dependent QS. Given the importance of cytochrome $bc_1$ for energy transduction and growth, we also compared aerobic growth and ATP production in the *P. aeruginosa* wild type with that of the *tat* and *petA* mutants. **S9 Fig** shows that there is a similar small reduction in growth for mutants compared

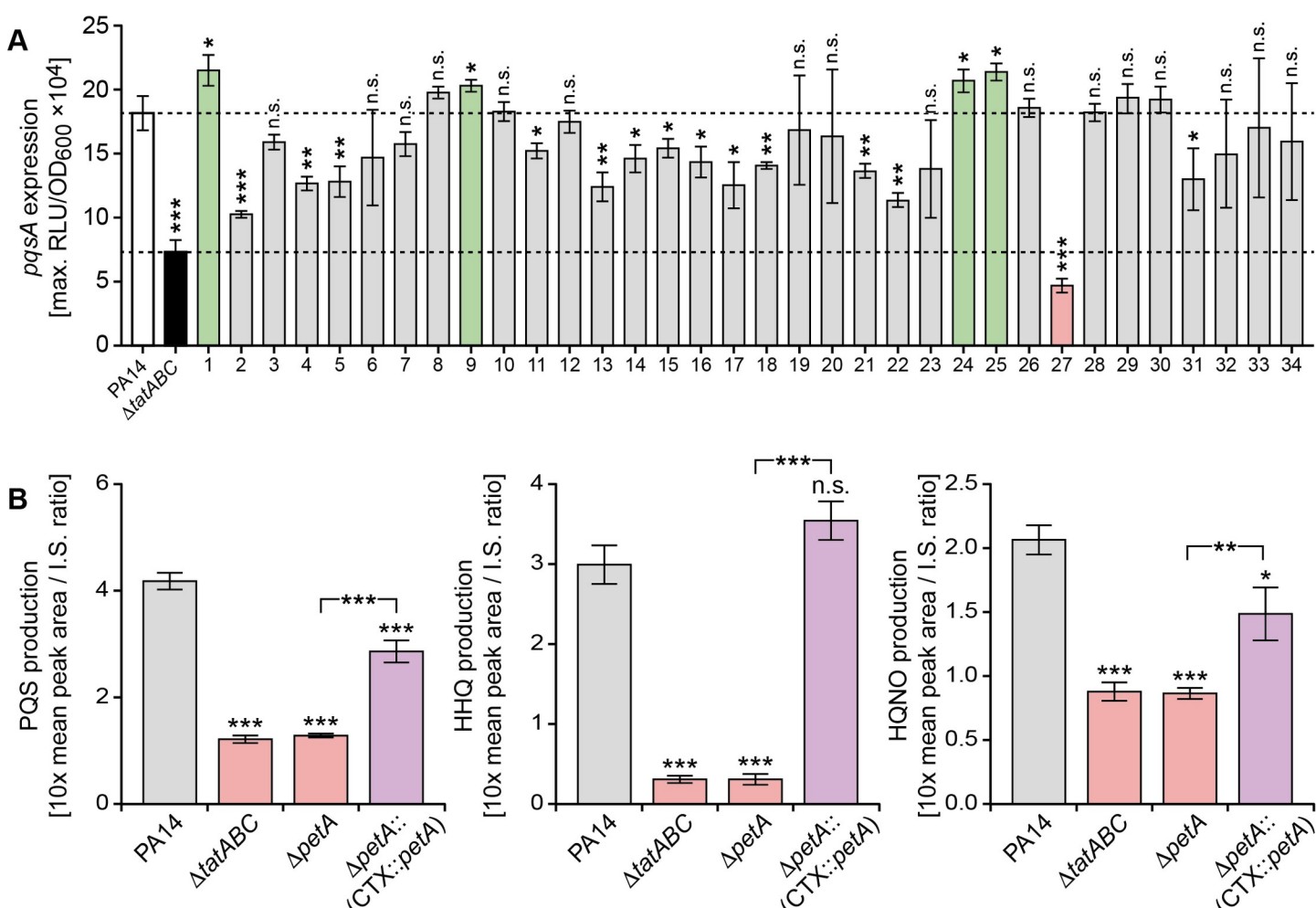

**Fig 10. Tat substrate screen for *P. aeruginosa* PA14 mutants with defects in *pqs* signaling uncovers a role for the cytochrome *bc₁* Rieske sub-unit.** (A) Comparison of peak *pqsA* expression in each of 34 Tat substrate mutants (**S3 Table**) transformed with a CTX::*pqsA'-lux* fusion compared with the PA14 wild type (white bar) and the Δ*tatABC* mutant (black bar). The bars represent mutants where *pqsA* expression is higher (green bar) or lower (grey bar) or the same (grey bar) as the wild type. PA14 mutant 27 (pink bar) has the lowest *pqsA* expression and carries a deletion in PA14-57570 (*petA*), the Rieske subunit of cytochrome *bc₁*. (B) Production of PQS, HHQ and HQNO after 16 h growth by *P. aeruginosa* PA14, the Δ*tatABC* and Δ*petA* mutants and the genetically complemented PA14 mutant (Δ*petA*::(CTX::*petA*)). ***$p < 0.001$, **$p < 0.01$, and *$p < 0.05$; n.s. not significant.

with the wild type. Consistent with these observations, ATP levels were reduced by ~35% in the Δ*tat* and Δ*petA* mutants respectively compared with wild type (**S10 Fig**). This finding suggested that in *P. aeruginosa* strains carrying mutations in *tat* or cytochrome bc₁ the energy required to synthesize secondary metabolites such as the AQs is diverted to maintain primary metabolism.

## Discussion

To gain further insights into eDNA release, a *P. aeruginosa* Tn mutant library was screened and two groups of mutants exhibiting reduced eDNA release were identified. The first of these contained Tn insertions in *pqsC*, *pqsH* and *pqsR*. Mutations in *pqsC* or *pqsR* in common with *pqsA* both abrogate AQ biosynthesis while *pqsH* mutants are unable to produce 3-hydroxy-AQs such as PQS but maintain production of AQs and AQ N-oxides such as HHQ and HQNO respectively [17, 54]. These data suggest that eDNA release is likely to require

PQS/HHQ rather than the effector protein PqsE. This is because although *pqsE* mutants in common with *pqsA* mutants form poor biofilms, *pqsE pqsA* double mutants could not be complemented by *pqsE* alone to restore biofilm development [26]. This contrasts with pyocyanin and lectin A for example, as production of both can be restored by PqsE in the absence of AQ biosynthesis [26]

The Tn insertions in the second group of *P. aeruginosa* eDNA release mutants were in *tatA* and *tatB* [42,43]. In *P. aeruginosa*, proteins exported by the Tat system are most frequently terminally localized in the periplasm, but some (e.g. phospholipase C) can be transported across the outer membrane by the Xcp type II secretion system, thereby becoming extracellular [43]. Using prediction algorithms for Tat substrates, 44 putative *P. aeruginosa* PA14 Tat signal peptides have been identified and of these, 34 confirmed experimentally [45]. These include phospholipases, and proteins involved in pyoverdine-mediated iron-uptake, respiration, osmotic stress defence, motility, and biofilm formation [44, 45]. However, none of these are known to be involved in AQ biosynthesis, transport or PQS signal transduction.

The formation of flat, eDNA deficient, tobramycin-sensitive biofilms and the reduction in rhamnolipid, pyocyanin and MV production by the *P. aeruginosa tatA* mutant are consistent with the *pqsA* mutant biofilm phenotype [8] and implied the existence of a link between Tat export and PQS signalling. Since transcriptomic studies of PQS signalling in *P. aeruginosa* have not provided any evidence that the Tat export system is QS controlled [17,26], we considered it likely that mutation of the *tat* genes resulted in perturbed PQS signalling. This observation could also account, at least in part, for the reduced virulence of *P. aeruginosa tat* mutants in a rat pulmonary infection model [44]. For both *P. aeruginosa* PAO1 and PA14 strains *pqsA* expression and AQ production were reduced in the respective *tat* mutants while exogenous provision of either PQS or HHQ failed to fully restore *pqsA* expression in a *tatA ΔpqsA* mutant. The consequences of perturbed PQS signalling are clearly apparent in the altered expression profiles of the *rhlA'-lux* and *phzA1'-lux* fusions. Thus disruption of Tat clearly impairs the ability of *P. aeruginosa* either to fully induce PQS-dependent QS or to respond fully to exogenous PQS.

Since pyocyanin can be produced in the absence of PQS by ectopic expression of *pqsE* [26], it was possible that the *tat* phenotype was either a consequence of the inability of PQS/HHQ to activate the *pqsABCDE* operon via PqsR to generate sufficient PqsE protein or because the activity of PqsE depends on a functional Tat system. However, introduction of a plasmid-borne copy of *pqsE* into the *tat* mutants fully restored pyocyanin production suggesting that the *tat* mutant phenotype is not due to the inability of PqsE to function but rather a failure of the PQS auto-induction circuitry to produce sufficient PQS/HHQ to efficiently activate *pqsABCDE*.

To uncover a direct link between Tat and PQS signalling, we screened a bank of Tat substrate mutants for reduced *pqsA* expression and identified PetA as primarily responsible for the PQS signalling defect. As PetA is one of the three protein sub-units that constitute the cytochrome $bc_1$ complex, we also showed that a similar PQS signalling defect was apparent in *cytB* or $cytC_1$ mutants. Since growth of both the *tat* and cytochrome $bc_1$ subunit mutants was reduced compared with the wild type (**S9 Fig**), our working hypothesis is that shutting down secondary metabolism and virulence factor production would aid conservation of energy for primary metabolism and growth. This shutdown could be efficiently achieved by preventing PQS autoinduction since it co-ordinates the expression of multiple genes involved in secondary metabolite production [17] including the biosynthesis of over 50 different AQs [54] that are heavily-dependent on tryptophan/anthranilate and fatty acid metabolic pathways. How PQS 'shutdown' is achieved at the molecular level in the *tat* mutants is not yet apparent but given that *tat ΔpqsA* double mutants do not respond to exogenous PQS, it may involve uptake

into the cell and/or access to the response regulator PqsR(MvfR). At present there is little information on the mechanism by which PQS gains intracellular access although PQS is known to induce MV formation via a direct interaction with LPS and to be trafficked within MVs [33,34]. This is consistent with the reduction in MV production that we observed in the *tat* and *pqsA* mutants.

As shown in **Figs 1** and **S1**, PQS biosynthesis generates a diverse series of AQs including the *N*-oxide, HQNO, a well known cytochrome $bc_1$ inhibitor [57]. HQNO binds to the quinone reduction (Qi) site of the cytochrome $bc_1$ complex disrupting the flow of electrons resulting in the leakage of reducing equivalents to oxygen [57]. This results in the generation of reactive oxygen species that cause *P. aeruginosa* cell death and autolysis favouring biofilm formation and antibiotic tolerance [57]. Rieske subunit and cytochrome $bc_1$ mutants do not undergo autolysis and are insensitive to exogenous HQNO [57]. Similarly, *P. aeruginosa pqsL* mutants that are unable to produce HQNO also fail to undergo autolysis [57]. Our data also show that in common with the *tat* mutants, the *petA* as well as *cytB* and $cytC_1$ mutants all produce lower levels of HQNO. This suggests that lack of eDNA in the *tat* mutant biofilms is because the cells do not undergo limited autolysis [57] as they lack the self-poisoning mechanism that depends on both an intact cytochrome $bc_1$ and sufficient HQNO [57]. Thus, lack of the Rieske sub-unit export is primarily responsible for the Tat-mediated perturbation of PQS-dependent QS, the loss of virulence factor production, biofilm eDNA and the tobramycin tolerance of *P. aeruginosa* biofilms. Given the importance of PQS signalling and the Tat system to virulence and biofilm maturation in *P. aeruginosa*, our findings underline the potential of the Tat system as a drug target for novel antimicrobial agents.

## Materials and methods

### Bacterial strains and growth conditions

The *P. aeruginosa* and *E. coli* strains used are listed in **S1 Table** and were grown in LB or ABTG [8] at 37˚C unless otherwise stated. *P. aeruginosa* biofilms were cultivated at 30˚C in flow-chambers irrigated with FAB medium [58] supplemented with 0.3 mM glucose or in M9 medium with succinate (for static biofilms). Selective media were supplemented with ampicillin (Ap; 100 µg ml$^{-1}$), gentamicin (Gm; 60 µg ml$^{-1}$), or streptomycin (Sm; 100 µg ml$^{-1}$). PQS, HHQ and 2-heptyl-4-hydroxyquinoline N-oxide (HQNO) were synthesized and characterized in house as described before [30, 59] and dissolved in methanol before being added to growth media at the appropriate concentration.

### Mutant construction, screening and validation

A *P. aeruginosa* PAO1 Tn mutant library was constructed using the Mariner Tn vector pBT20 as previously described [60]. Transconjugants carrying Tn insertions were picked from the selective plates and inoculated into microtiter plates containing ABTG medium [61] supplemented with propidium iodide (PI) and the level of red fluorescence quantified. Mutants producing reduced levels of eDNA were selected and the sequences flanking the Tn insertion identified by arbitrary PCR essentially as described by Friedman and Kolter [62] but using the specific primers TnM1 and TnM2 (**S2 Table**). DNA Sequencing was performed by Macrogen, Seoul, Korea with primer TnMseq (**S2 Table**). An in-frame *P. aeruginosa tatABC* deletion mutant was generated by allelic exchange using the oligonucleotide primers Tat3D-UF, Tat3-D-UR, Tat3D-DF and Tat3D-DR (**S2 Table**) to introduce the up- and down- stream regions of the *tatABC* locus into the suicide vector pME3087 to generate pME3087::*tatABC* (**S1 Table**). The latter was introduced into *P. aeruginosa* via conjugation with *E. coli* S17-1 λpir followed by enrichment for tetracycline-sensitive cells as described by Ye *et al* [63]. The Δ*tatABC*

deletion in *P. aeruginosa* PAO1 was confirmed by PCR and sequence analysis. To generate the PA14 Δ*petA* mutant, 500 bp upstream and downstream of the gene were amplified using respectively primers S.PA4431upFor/S.PA4431upRev and S.PA4431downFor/S.PA4431down-Rev listed in **S2 Table**. The PCR product was cloned in pKNG101 suicide vector by one-step sequence and ligation-independent cloning (SLIC) [64] and sequenced. The resulting plasmid, pKNGΔ*petA*, maintained in the *E. coli* CC118λpir strain, was then mobilized in *P. aeruginosa* strains. The mutant, in which the double recombination events occurred, was confirmed by PCR analysis. A similar strategy was used to construct allelic replacement mutants for each of 34 validated Tat substrates as well as the *cytB* and *cytC*$_1$ mutants [45]. These strains are summarized in **S3 Table** and their validation will be described in detail elsewhere. For the generation of the *petA* cis-complemented strain, PA14Δ*petA*::*(CTX1::petA)*, the *petA* genes along with a 500 bp fragment corresponding to the putative promoter region for the *petA* gene were PCR amplified using S-4431CTXFor/S-4431CTXRev and cloned by SLIC into the mini-CTX1 vector yielding pminiCTX1-*petA*. Transfer of this plasmid in *P. aeruginosa* Δ*petA* strain was carried out by triparental mating using the conjugative properties of the helper plasmid pRK2013. The recombinant clones containing the mini-CTX inserted at the *attB* locus on the *P. aeruginosa* genome were selected on tetracycline-containing PIA generating PA14Δ*petA*::*(attB::petA)*.

## Construction of a *tatA* complementation plasmid

The *tatA* gene was amplified by PCR using primers F*tatA* and R*tatA* (**S2 Table**), introduced into pUCP22 (**S1 Table**) and electroporated into the *P. aeruginosa tatA* and *tatA* Δ*pqsA* mutants. Transformants were selected on LB plates containing 200 μg ml$^{-1}$ carbenicillin.

## Bioluminescent reporter assays

To investigate the impact of the *tat* mutation and Tat inhibitor Bayer 11–7082 [48] on PQS signalling, transcriptional fusions between the promoter regions of *pqsA*, *pqsR*, *rhlA*, *phzA1 and phzA2* and the *luxCDABE* operon were constructed using the miniCTX-*lux* plasmid as previously described [17]. In addition, a constitutively bioluminescent reporter using a miniCTX::*tac-luxCDABE* promoter fusion was constructed as a control for Bayer 11–7082. Bioluminescence as a function of bacterial growth was quantified in 96 well plates using a combined luminometer- spectrometer.

Semi-quantification of cellular ATP was carried out using the BacTiter-Glo$^{TM}$ Microbial Cell Viability Assay (Promega). Briefly, the *P. aeruginosa* PA14, Δ*tatABC* mutant and Δ*petA* mutant were grown in LB broth for 8 h, diluted 1000-fold with fresh media and mixed with equal volume of BacTiter-Glo$^{Tm}$ reagent in a 96-well plate. After a 5 min incubation period, luminescence for each well was recorded in an automated plate reader.

## Cultivation of biofilms

Biofilms were grown in flow-chambers with individual channel dimensions of 1 x 4 x 40 mm as described previously [65]. One hour after inoculation, with bacteria, the growth medium flow (0.2 mm/s corresponding to laminar flow with a Reynolds number of 0.02) was started. When required, eDNA in biofilms was stained with 1 μM ethidium bromide prior to microscopy. Tobramycin (10 μM) was added to the biofilm medium after 4 days of cultivation. After 24 h of tobramycin treatment, propidium iodide (10 μM) was added to the flow cells to visualize the dead cells via confocal laser scanning microscopy. *P. aeruginosa* PA14 and *petA* mutant biofilms were grown under static conditions over 48 h at 37˚C on glass slides (Ibidi)

incorporating 300 μl chambers. After 48 h incubation, spent medium was removed and the biofilm eDNA stained with YoYo-1 (40 μM).

### Microscopy and image processing of flow cell biofilms

All images of flow-chamber-grown and static biofilms were captured with a confocal laser scanning microscope (CLSM) equipped with detectors and filter sets for monitoring green fluorescent protein, Syto9, propidium iodide, and ethidium bromide. Images were obtained using a 63x/1.4 objective or a 40x/1.3 objective. Simulated 3-D images and sections were generated using the IMARIS software package (Bitplane AG, Zürich, Switzerland).

### AQ, pyocyanin, rhamnolipid and MV analysis

The AQs (PQS, HHQ and HQNO) were quantified by LC-MS/MS after extracting cell free supernatants or whole bacterial cell cultures in triplicate with acidified ethyl acetate or methanol respectively as described by Ortori *et al* [59]. Pyocyanin was extracted with chloroform and quantified spectrophotometrically [26]. Rhamnolipids were quantified indirectly using the orcinol method [35]. For PQS solubilization experiments, rhamnolipids were purified as described by Muller *et al* [66]. MVs were harvested by ultracentrifugation, the pellets resuspended in 10mM HEPES buffer and the lipid content quantified using FM4-64 essentially as described previously [67]. MV production was normalized by dividing the lipid fluorescence units by CFU values determined by dilution plating. Assays were performed in triplicate at least twice.

### Statistical analysis

Significance for differences between wild type and isogenic mutants was determined by two-tailed *t*-tests where ****$p < 0.001$, ***$p < 0.001$, **$p < 0.01$, and *$p < 0.05$ and n.s., not significant.

### Supporting information

**S1 Fig. Biochemical basis of AQ biosynthesis.** PqsA catalyses the formation of anthraniloyl-CoA that is condensed with malonyl-CoA by PqsD to form 2-aminobenzoylacetyl-CoA (2-ABA-CoA). The latter is converted to 2-aminobenzoylacetate (2-ABA) via the thioesterase functionality of PqsE. The PqsBC heterodimer condenses 2-ABA with octanoyl-CoA to generate HHQ. PQS is formed through the oxidation of HHQ by PqsH. For AQ *N*-oxides such as 2-heptyl-4-hydroxyquinoline *N*-oxide (HQNO), 2-ABA is oxidized to 2-HABA by the alternative mono-oxygenase PqsL and then condensed with octanoyl-CoA by PqsBC to form HQNO. PqsBC can accept acyl-CoAs of different acyl chain lengths to generate diverse AQs and AQ *N*-oxides.
(TIF)

**S2 Fig. The Tat inhibitor Bayer 11–7082 has no effect on light output at 20 or 40 μM from a constitutive *P. aeruginosa* CTX::*tac'-luxCDABE* chromosomal reporter fusion.** The co-solvent DMSO, had no effect at 0.4 or 0.8% on the *lux* reporter fusion Data are presented as maximal light output as a function of growth (RLU/OD$_{600}$). Experiments were repeated in triplicate at least twice.
(TIF)

**S3 Fig. Deletion of the *tatABC* genes does not influence *pqsR* expression.** The data show that there are no differences in the expression of a chromosomal CTX::*pqsR'-luxCDABE* fusion

in the *P. aeruginosa* wild type compared with the Δ*tatABC* mutant Data are presented as maximal light output as a function of growth (RLU/OD$_{600}$). Experiments were repeated in triplicate at least twice.
(TIF)

**S4 Fig. HHQ and PQS do not accumulate intracellularly in a *tatA* Δ*pqsA* double mutant harboring the plasmid-borne *pqsABCD* genes in the absence of autoinduction.** Semi-quantitative analysis by LC-MS/MS of PQS (**A**) and HHQ (**B**) extracted with methanol from whole cell cultures of *P. aeruginosa* wild type and the *tatA* Δ*pqsA* mutant without (control) or with (+*pqsABCD*) the *pqsABCD* biosynthetic genes provided via pBBR1MCS-5::*pqsABCD* and harvested at 8 h and 16 h respectively. Experiments were repeated in triplicate.
(TIF)

**S5 Fig. Exogenous rhamnolipids do not enhance PQS-dependent expression of *pqsA* in a *tatA* Δ*pqsA* mutant. PQS (40 µM) was added with or without purified rhamnolipids (50 µg/ml) to a *pqsA* mutant (grey bars) or a *tatA* Δ*pqsA* mutant (pink bars) carrying chromosomal *pqsA'-lux* fusions.** Maximal light output as a function of growth (RLU/OD$_{600}$) is presented. Experiments were repeated in triplicate at least twice.
(TIF)

**S6 Fig. Rhamnolipid biosynthesis gene *rhlA* shows altered expression profiles in *P. aeruginosa* PA14 Δ*tatABC* and Δ*petA* mutants compared with wild type and fail to respond to exogenous PQS.** Bioluminescence from a chromosomal *rhlA'-lux* fusion as a function of growth (RLU/OD) over time when introduced into (**A**) the PA14 wild type, (**B**) Δ*tatABC* and (**C**) Δ*petA* mutants in the absence or presence of exogenous PQS (20 µM).
(TIF)

**S7 Fig. Deletion of *petA* in *P. aeruginosa* PA14 results in a reduction in the eDNA content of biofilms.** Biofilms of wild type PA14, the Δ*tatABC* and Δ*petA* mutants and the genetically complemented PA14 mutant (Δ*petA*::(CTX::*petA*)) were grown cultured statically and stained for eDNA with YOYO-1. (**A**) confocal fluorescence microscopy images and (**B**) eDNA quantification. Experiments were repeated in triplicate at least twice. ***p < 0.001, **p < 0.01.
(TIF)

**S8 Fig. Comparison of *pqsA* expression in *P. aeruginosa* PAO1 wild type, Δ*tatABC* and Δ*petA* mutants.** Experiments were repeated in triplicate at least twice. ***p < 0.001, **p < 0.01, and *p < 0.05; n.s. not significant.
(TIF)

**S9 Fig. Growth, *pqsA* expression and AQ production in wild type *P. aeruginosa* PA14 compared with Δ*cytB* (PA4429), Δ*cytC$_1$* (PA4430) Δ*petA* and Δ*tatABC* mutants.** ***p < 0.001, **p < 0.01, and *p < 0.05; n.s. not significant.
(TIF)

**S10 Fig. Comparison of cellular ATP levels in *P. aeruginosa* PA14 wild type, Δ*tatABC* and Δ*petA*.** Experiments were repeated in triplicate at least twice.
(TIF)

**S1 Table. Strains and plasmids used in this study.**
(DOCX)

**S2 Table. Oligonucleotide primers used in this study.**
(DOCX)

**S3 Table. *P. aeruginosa* PA14 Tat substrate mutants used in this study.**
(DOCX)

## Acknowledgments

The authors wish to thank Alex Truman for synthesis of the AQ and AQ-N-oxide standards.

## Author Contributions

**Conceptualization:** Liang Yang, Sophie Bleves, Miguel Cámara, Michael Givskov, Kim R. Hardie, Tim Tolker-Nielsen, Bérengère Ize, Paul Williams.

**Data curation:** Eliza Ye-Chen Soh, Frances Smith, Maxime Rémi Gimenez, Rebecca Munk Vejborg, Matthew Fletcher, Nigel Halliday.

**Formal analysis:** Eliza Ye-Chen Soh, Frances Smith, Maxime Rémi Gimenez, Liang Yang, Rebecca Munk Vejborg, Matthew Fletcher, Nigel Halliday, Stephan Heeb, Tim Tolker-Nielsen, Bérengère Ize, Paul Williams.

**Funding acquisition:** Sophie Bleves, Michael Givskov, Tim Tolker-Nielsen, Bérengère Ize, Paul Williams.

**Investigation:** Eliza Ye-Chen Soh, Frances Smith, Maxime Rémi Gimenez, Liang Yang, Rebecca Munk Vejborg, Matthew Fletcher, Nigel Halliday, Bérengère Ize.

**Methodology:** Eliza Ye-Chen Soh, Frances Smith, Maxime Rémi Gimenez, Matthew Fletcher, Nigel Halliday.

**Project administration:** Sophie Bleves, Michael Givskov, Tim Tolker-Nielsen, Bérengère Ize, Paul Williams.

**Resources:** Sophie Bleves.

**Supervision:** Sophie Bleves, Stephan Heeb, Kim R. Hardie, Tim Tolker-Nielsen, Bérengère Ize, Paul Williams.

**Validation:** Frances Smith.

**Visualization:** Stephan Heeb.

**Writing – original draft:** Paul Williams.

**Writing – review & editing:** Eliza Ye-Chen Soh, Frances Smith, Maxime Rémi Gimenez, Liang Yang, Rebecca Munk Vejborg, Matthew Fletcher, Nigel Halliday, Sophie Bleves, Stephan Heeb, Miguel Cámara, Michael Givskov, Kim R. Hardie, Tim Tolker-Nielsen, Bérengère Ize.

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
