## [Decision Letter · Decision Letter 0]

29 Mar 2021

Dear Prof. Williams,

Thank you very much for submitting your manuscript "Disruption of the Pseudomonas aeruginosa Tat system perturbs PQS-dependent quorum sensing and biofilm maturation through loss of the Rieske cytochrome bc1 sub-unit" for consideration at PLOS Pathogens. As with all papers reviewed by the journal, your manuscript was reviewed by members of the editorial board and by several independent reviewers. In light of the reviews (below this email), we would like to invite the resubmission of a significantly-revised version that takes into account the reviewers' comments.

We cannot make any decision about publication until we have seen the revised manuscript and your response to the reviewers' comments. Your revised manuscript is also likely to be sent to reviewers for further evaluation.

Sincerely,

Matthew Parsek, PhD

Associate Editor

PLOS Pathogens

Alan Hauser

Section Editor

PLOS Pathogens

Kasturi Haldar

Editor-in-Chief

PLOS Pathogens

orcid.org/0000-0001-5065-158X

Michael Malim

Editor-in-Chief

PLOS Pathogens

orcid.org/0000-0002-7699-2064

Reviewer's Responses to Questions

**Part I - Summary**

Reviewer #1: This study describes a screen for mutants defective in extracellular DNA release in Pseudomonas aeruginosa. The authors identify insertions in the tatA and tatB genes of PAO1 that show reduced levels of eDNA. The phenotype is linked to reduced PQS function and cannot be chemically complemented by exogenous PQS. Analysis of a library of PA14 carrying individual gene deletions for Tat substrate proteins identified that deletion of petA, encoding the Rieske iron-sulfur protein component of the cytochrome bc1 complex largely phenocopied the defective PQS signalling seen with the tat mutant strain.

This is an interesting study with a large body of work that is in general well performed. The link between PetA, Tat and PQS is clearly made, however at present there is no molecular explanation for the observed phenotypes, leaving the reader unclear about how PetA affects PQS signalling. The authors need to come up with a model for the role of PetA (and the bc1 complex) to account for their findings.

Reviewer #2: In this manuscript they are trying to understand the mechanism behind eDNA release in P. aeruginsoa. To do this, 10,000 transposon mutants were screened with propidium iodide which has increased fluorescence upon DNA binding. Strains with less eDNA were rescreened and transposon insertion sites were determined. 34 mutants were obtained and many had insertions in pqsC or pqsH, which indicates that PQS was important for eDNA release, as expected. They also found mutations in tatA and tatB, the twin arginine translocation locus, which exports folded proteins that have Arg-Arg in the signal sequence. They found that PQS controlled factors are decreased in the tatA mutant and that PQS is down about 50%. In addition, a TatA inhibitor caused a large decrease in pqsA expression. PQS production in the tatA pqsA mutant can not be complemented by expressing pqsABCD and PQS activity was not complemented by exogenous PQS in a tatA mutant.

So they had a link between TatA and PQS but the question was how did it work? To figure this out they looked at the TatA exported products. P. aeruginosa has at least 34 Tat secreted products and they tested mutants in all 34 for pqsA expression to find a product that affected PQS signalling. PA4431 encodes PetA, the Rieske subunit of cytochrome bc1 complex and this mutant had PQS production decreased similar to that of the tatA mutant. Of interest - HQNO (a quinolone from the pqsABCD operon) poisons cytochrome bc1 and Rieske subunit mutants do not undergo autolyis and are insensitive to HQNO. This led them to conclude that due to decreased quinolone production, the tatA mutant does not undergo the limited autolysis that occurs with HQNO present and therefore does not have as much eDNA around to form biofilms. It’s an interesting manuscript and it had a finding that kind of surprised me. The writing could use a bit more “storytelling” as there were times that I wasn’t sure where they were going.

Reviewer #3: This is a well-written manuscript on an interesting and relevant topic. The conclusions are supported by an extensive data set and all experiments appear to have been carried out according to the state-of-the-art. I have however two major comments.

**Part II – Major Issues: Key Experiments Required for Acceptance**

Reviewer #1: Is this phenotype unique to the loss of PetA? Or is it phenocopied by deletion of genes encoding the other components of the bc1 complex? It would be important to test this.

P. aeruginosa has other terminal oxidases that bypass the bc1 complex and take electrons directly from the quinone pool (e.g. CYO). If one of these is (over)expressed in the tat or petA mutant can the PQS defect be rescued?

PQS and HHQ are chemically very similar to ubiquinone, and in particular to menaquinone. Is it possible that one or both of these molecules interact with the bc1 complex and are chemically modified in some way (for example by oxidation or reduction), and that this generates the active signalling molecule? The lack of exogenous complementation of the tat mutant by PQS is very telling in this regard and would be consistent with a role for the bc1 complex in generating the active form of PQS/HHQ. The other alternative for the lack of chemical complementation is that uptake of PQS is abolished (but given that overexpression of pqsABCD did not complement the tat mutant either this seems less likely). Is the protonmotive force affected in the tat/petA mutant strains?

Reviewer #2: -

Reviewer #3: First of all, it is unclear why most of the work is done in strain PAO1, while strain PA14 was chosen to make the 34 allelic replacement mutants. Isn’t there a PAO1 petA mutant? If not, why not construct it and then confirm that PetA is the Tat substrate that is responsible for perturbed PQS signaling, also in PAOA.

Secondly, a weakness of the study is that for the biofilm work only flow-cell grown biofilms are considered. This model has without any doubt its value, but at the same time it is also the only model in which we can observe ‘mushroom-shaped’ biofilms. I believe this study would be much stronger if its key findings could be confirmed in more in vivo-like conditions (e.g. biofilms formed in growth media resembling CF sputum or the wound environment).

**Part III – Minor Issues: Editorial and Data Presentation Modifications**

Reviewer #1: Important details of the mariner Tn library are missing. There is no information on: 1) the size of the library (how many actual clones were produced), either before or after eliminating the double insertions; 2) how the double insertions were found; 3) how many clones passed selection and were identified as poor eDNA releasers; 4) the exact identity and frequency of all insertions found. How many of the 34 insertions were in pqs genes? This section is vaguely written and needs to be clarified. Can the authors comment on the reasons why no insertion in either tatC or petA were found? While this does not invalidate the authors’ subsequent investigation of the role of Tat/PetA, it may reflect incomplete coverage of the library, and this should be discussed.

The title of the manuscript is not strictly accurate as the Rieske subunit is not lost in the tat mutant but mislocalized. Activity is lost.

Please add anthranilate to Fig 1 as the precursor for HHQ/PQS biosynthesis

Do the tat or petA mutants grow more slowly than the wild type strains? This would be useful information to provide as it might impinge on some of the findings.

The small molecule inhibitor Tat data does not add anything and should be removed. It should be noted that Bageshwar et al (PLoS ONE 2016 https://doi.org/10.1371/journal.pone.0149659) concluded that although Bayer-11 7082 inhibited growth under Tat++ conditions it did not affect Tat transport in vitro.

It is clear that PetA alone has major effect in biofilm development and associated events. However, of the other 30+ Tat substrates many have small but, presumably, cumulative contributions, so they authors should acknowledge this by clarifying that PetA is the major factor rather than the sole factor affecting biofilm formation.

The discussion is very long and to in places is repetitious of the results section. It should be shortened. The authors speculate on the role of PetA in leading to cell bursts and eDNA release – what is the evidence that ROS production is the means, by which PetA contributes to these events?

Reviewer #2: Comments.

1. Line 225. A line should be added to explain the significance of a tatA mutant being more sensitive to tobramycin.

2. Figure S3 should be a regular figure as it is important.

3. Line 276. There should be a line of discussion for this result. Also – same issue at line 293.

4. The discussion has a lot of restating of results.

5. The meaning of the tatA mutant isn’t touched upon until line 481. It should be explained much earlier and reiterated in the section at line 481.

Reviewer #3: Finally, while reading the manuscript it felt like there was a lot of repetition in the Discussion and I would suggest to consider combining Results and Discussion.

PLOS authors have the option to publish the peer review history of their article (what does this mean?). If published, this will include your full peer review and any attached files.

Reviewer #1: No

Reviewer #2: No

Reviewer #3: No
---

## [Decision Letter · Decision Letter 1]

10 Aug 2021

Dear Prof. Williams,

We are pleased to inform you that your manuscript 'Disruption of the Pseudomonas aeruginosa Tat system perturbs PQS-dependent quorum sensing and biofilm maturation through lack of the Rieske cytochrome bc1 sub-unit' has been provisionally accepted for publication in PLOS Pathogens.

Best regards,

Matthew Parsek, PhD

Associate Editor

PLOS Pathogens

Alan Hauser

Section Editor

PLOS Pathogens

Kasturi Haldar

Editor-in-Chief

PLOS Pathogens

orcid.org/0000-0001-5065-158X

Michael Malim

Editor-in-Chief

PLOS Pathogens

orcid.org/0000-0002-7699-2064

Reviewer Comments (if any, and for reference):

Reviewer's Responses to Questions

**Part I - Summary**

Reviewer #1: The revised version of the manuscript has addressed my concerns

Reviewer #3: (No Response)

**Part II – Major Issues: Key Experiments Required for Acceptance**

Reviewer #1: None

Reviewer #3: (No Response)

**Part III – Minor Issues: Editorial and Data Presentation Modifications**

Reviewer #1: None

Reviewer #3: (No Response)

PLOS authors have the option to publish the peer review history of their article (what does this mean?). If published, this will include your full peer review and any attached files.

Reviewer #1: No

Reviewer #3: No

---

## [Editor Report · Acceptance letter]

25 Aug 2021

Dear Prof. Williams,

We are delighted to inform you that your manuscript, "Disruption of the Pseudomonas aeruginosa Tat system perturbs PQS-dependent quorum sensing and biofilm maturation through lack of the Rieske cytochrome bc1 sub-unit," has been formally accepted for publication in PLOS Pathogens.

Best regards,

Kasturi Haldar

Editor-in-Chief

PLOS Pathogens

orcid.org/0000-0001-5065-158X

Michael Malim

Editor-in-Chief

PLOS Pathogens

orcid.org/0000-0002-7699-2064